# Deep Neural Networks with Inexact Matching for Person Re-Identification

**Arulkumar Subramaniam**
Indian Institute of Technology Madras
Chennai, India 600036
aruls@cse.iitm.ac.in

**Moitreya Chatterjee**
Indian Institute of Technology Madras
Chennai, India 600036
metro.smiles@gmail.com

**Anurag Mittal**
Indian Institute of Technology Madras
Chennai, India 600036
amittal@cse.iitm.ac.in

## Abstract

Person Re-Identification is the task of matching images of a person across multiple camera views. Almost all prior approaches address this challenge by attempting to learn the possible transformations that relate the different views of a person from a training corpora. Then, they utilize these transformation patterns for matching a query image to those in a gallery image bank at test time. This necessitates learning good feature representations of the images and having a robust feature matching technique. Deep learning approaches, such as Convolutional Neural Networks (CNN), simultaneously do both and have shown great promise recently.

In this work, we propose two CNN-based architectures for Person Re-Identification. In the first, given a pair of images, we extract feature maps from these images via multiple stages of convolution and pooling. A novel inexact matching technique then matches pixels in the first representation with those of the second. Furthermore, we search across a wider region in the second representation for matching. Our novel matching technique allows us to tackle the challenges posed by large viewpoint variations, illumination changes or partial occlusions. Our approach shows a promising performance and requires only about half the parameters as a current state-of-the-art technique. Nonetheless, it also suffers from false matches at times. In order to mitigate this issue, we propose a fused architecture that combines our inexact matching pipeline with a state-of-the-art exact matching technique. We observe substantial gains with the fused model over the current state-of-the-art on multiple challenging datasets of varying sizes, with gains of up to about 21%.

## 1 Introduction

Successful object recognition systems, such as Convolutional Neural Networks (CNN), extract "distinctive patterns" that describe an object (e.g. a human) in an image, when "shown" several images known to contain that object, exploiting Machine Learning techniques [1]. Through successive stages of convolutions and a host of non-linear operations such as pooling, non-linear activation, etc., CNNs extract complex yet discriminative representation of objects that are then classified into categories using a classifier, such as softmax.

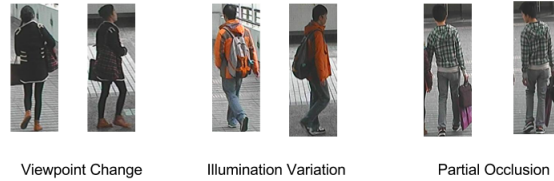

Viewpoint Change       Illumination Variation       Partial Occlusion

Figure 1: Some common challenges in Person Re-Identification.

## 1.1 Problem Definition

One of the key subproblems of the generic object recognition task is recognizing people. Of special interest to the surveillance and Human-Computer interaction (HCI) community is the task of identifying a particular person across multiple images captured from the same/different cameras, from different viewpoints, at the same/different points in time. This task is also known as *Person Re-Identification*. Given a pair of images, such systems should be able to decipher if both of them contain the same person or not. This is a challenging task, since appearances of a person across images can be very different due to large viewpoint changes, partial occlusions, illumination variations, etc. Figure 1 highlights some of these challenges. This also leads to a fundamental research question: Can CNN-based approaches effectively handle the challenges related to Person Re-Identification?

CNN-based approaches have recently been applied to this task, with reasonable success [2, 3] and are amongst the most competitive approaches for this task. Inspired by such approaches, we explore a set of novel CNN-based architectures for this task. We treat the problem as a classification task. During training, for every pair of images, the model is told whether they are from the same person or not. At test time, the posterior classification probabilities obtained from the models are used to rank the images in a gallery image set in terms of their similarity to a query image (probe).

In this work, we propose two novel CNN-based schemes for Person Re-Identification. Our first model hinges on the key observation that due to a wide viewpoint variation, the task of finding a match between the pixels of a pair of images needs to be carried out over a larger region, since "matching pixels" on the object may have been displaced significantly. Secondly, illumination variations might cause the absolute intensities of image regions to vary, rendering exact matching approaches ineffective. Finally, coupling these two solutions might provide a recipe for taking care of partial occlusions as well. We call this first model of ours, Normalized X-Corr. However, the flexibility of inexact (soft) matching over a wider search space comes at the cost of occasional false matches. To remedy this, we propose a second CNN-based model which fuses a state-of-the-art exact matching technique [2] with Normalized X-Corr. We hypothesize that proper training allows the two components of the fused network to learn complimentary patterns from the data, thus aiding the final classification. Empirical results show Normalized X-Corr to hold promise and the Fused network outperforming all baseline approaches on multiple challenging datasets, with gains of upto 21% over the baselines.

In the next section, we touch upon relevant prior work. We present our methodology in Section 3. Sections 4 and 5 deal with the Experiments and the discussion of the obtained Results thereof. Finally, we conclude in Section 6, outlining some avenues worthy of exploration in the future.

## 2 Related Work

In broad terms, we categorize the prior work in this field into Non-Deep and Deep Learning approaches.

## 2.1 Non-Deep Learning Approaches

Person Re-Identification systems have two main components: Feature Extraction and Similarity Metric for matching. Traditional approaches for Person Re-Identification either proposed useful features, or discriminative similarity metrics for comparison, or both.

Typical features that have proven useful include color histograms and dense SIFT [4] features computed over local patches in the image [5, 6]. Farenzena et al. represent image patches by exploiting features that model appearance, chromatic content, etc [7]. Another interesting line of work on feature representation attempts to learn a bag-of-features (a.k.a. a dictionary)-based approach for image representation [8, 9]. Further, Prosser et al. show the effectiveness of learning a subspace for representing the data, modeled using a set of standard features [10]. While these approaches show promise, their performance is bounded by the ability to engineer good features. Our models, based on deep learning, overcome this handicap by learning a representation from the data.

There is also a substantial body of work that attempts to learn an effective similarity metric for comparing images [11, 12, 13, 14, 15, 16]. Here, the objective is to learn a distance measure that is indicative of the similarity of the images. The Mahalanobis distance has been the most common metric that has been adopted for matching in person re-identification [5, 17, 18, 19]. Some other metric learning approaches attempt to learn transformations, which when applied to the feature space, aligns similar images closer together [20]. Yet other successful metric learning approaches are an ensemble of multiple metrics [21]. In contrast to these approaches, we jointly learn both features and discriminative metric (using a classifier) in a deep learning framework.

Another interesting line of non-deep approaches for person re-identification have claimed novelty both in terms of features as well as matching metrics [22, 23]. Many of them rely on weighing the hand-engineered image features first, based on some measure such as saliency and then performing matching [6, 24, 25]. However, this is done in a non-deep framework unlike ours.

### 2.2 Deep Learning based Approaches

There has been relatively fewer prior work based on deep learning for addressing the challenge of Person Re-Identification. Most deep learning approaches exploit the CNN-framework for the task, i.e. they first extract highly non-linear representations from the images, then they compute some measure of similarity. Yi et al. propose a Siamese Network that takes as input the image pair that is to be compared, performs 3 stages of convolution on them (with the kernels sharing weights), and finally uses cosine similarity to judge their extent of match [26]. Both of our models differ by performing a novel inexact matching of the images after two stages of convolution and then processing the output of the matching layer to arrive at a decision.

Li et al. also adopt a two-input network architecture [3]. They take the product of the responses obtained right after the first set of convolutions corresponding to the two inputs and process its output to obtain a measure of similarity. Our models, on the other hand, are significantly deeper. Besides, Normalized X-Corr stands out by retaining the matching outcome corresponding to every candidate in the search space of a pixel rather than choosing only the maximum match. Ahmed et al. too propose a very promising architecture for Person Re-Identification [2]. Our models are inspired from their work and does incorporate some of their features, but we substantially differ from their approach by performing an inexact matching over a wider search space after two stages of convolution. Further, Normalized X-Corr has fewer parameters than Ahmed et al. [2].

Finally, our Fused model is a one of a kind deep learning architecture for Person Re-Identification. This is because a combination (fusion) of multiple deep frameworks has hitherto remained unexplored for this task.

## 3 Proposed Approach

### 3.1 Our Architecture

In this work, we propose two architectures for Person Re-Identification. Both of our architectures are a type of "Siamese"-CNN model which take as input two images for matching and outputs the likelihood that the two images contain the same person.

### 3.1.1 Normalized X-Corr

The following are the principal components of the Normalized X-Corr model, as shown in Fig 2.

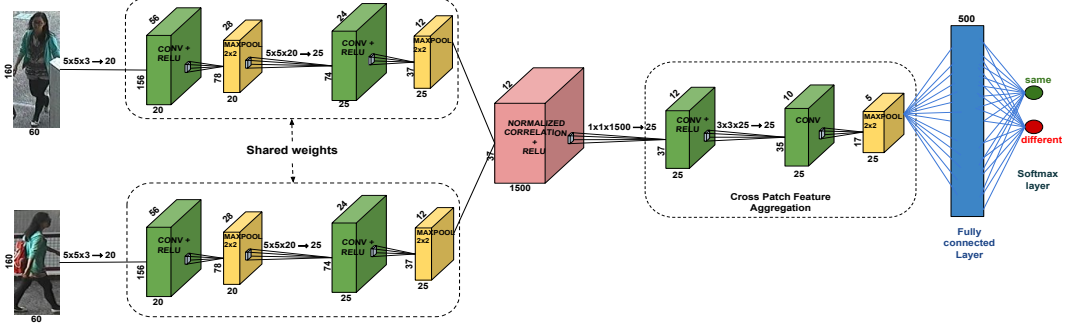

Figure 2: Architecture of the Normalized X-Corr Model.

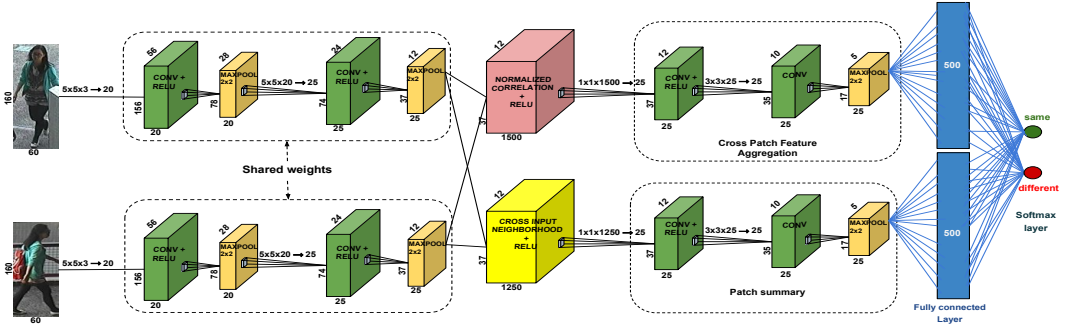

Figure 3: Architecture of the Fused Network Model.

**Tied Convolution Layers**: Convolutional features have been shown to be effective representation of images [1, 2]. In order to measure similarity between the input images, the applied transformations must be similar. Therefore, along the lines of Ahmed et al., we perform two stages of convolutions and pooling on both the input images by passing them through two input pipelines of a "Siamese" Network, that share weights [2]. The first convolution layer takes as input images of size $60 \times 160 \times 3$ and applies 20 learned filters of size $5 \times 5 \times 3$, while the second one applies 25 learned filters of size $5 \times 5 \times 20$. Both convolutions are followed by pooling layers, which reduce the output dimension by a factor of 2, and ReLU (Rectified Linear Unit) clipping. This gives us 25 maps of dimension $12 \times 37$ as output from each branch which are fed to the next layer.

**Normalized Correlation Layer**: This is the first layer that captures the similarity of the two input images; subsequent layers build on the output of this layer to finally arrive at a decision as to whether the two images are of the same person or not. Different from [2, 3], we incorporate both inexact matching and wider search. Given two corresponding input feature maps $X$ and $Y$, we compute the normalized correlation as follows. We start with every pixel of $X$ located at $(x, y)$, where x is along the width and y along the height (denoted as $X(x, y)$). We then create two matrices. The first is a $5 \times 5$ matrix representing the $5 \times 5$ neighborhood of $X(x, y)$, while the second is the corresponding $5 \times 5$ neighborhood of $Y$ centered at $(a, b)$, where $1 \leq a \leq 12$ and $y - 2 \leq b \leq y + 2$. Now, markedly different from Ahmed et al. [2], we perform inexact matching over a wider search space, by computing a *Normalized Correlation* between the two patch matrices. Given two matrices, $E$ and $F$, whose elements are arranged as two N-dimensional vectors, the Normalized Correlation is given by:

$$normxcorr(E, F) = \frac{\sum_{i=1}^{N}(E_i - \mu_E)(F_i - \mu_F)}{(N-1).\sigma_E.\sigma_F},$$

where $\mu_E, \mu_F$ denotes the means of the elements of the 2 matrices, E and F respectively, while $\sigma_E, \sigma_F$ denotes their respective standard deviations (a small $\epsilon = 0.01$ is added to $\sigma_E$ and $\sigma_F$ to avoid division by 0). Interestingly, Normalized Correlation being symmetric, we need to model the computation only in one-way, thereby cutting down the number of parameters in subsequent layers.

For every pair of $5 \times 5$ matrices corresponding to a given pixel in image $X$, we arrange the normalized correlation values in different feature maps. These feature maps preserve the spatial ordering of

pixels in $X$ and are also $12 \times 37$ each, but their pixel values represent normalized correlation. This gives us 60 feature maps of dimension $12 \times 37$ each. Now similarly, we perform the same operation for all 25 pairs of maps that are input to the Normalized Correlation layer, to obtain an output of 1500, $12 \times 37$ maps. One subtle but important difference between our approach and that of Li et al. [3] is that we preserve every correlation output corresponding to the search space of a pixel, $X(x, y)$, while they only keep the maximum response. We then pass these set of feature maps through a ReLU, to discard probable noisy matches.

The mean subtraction and standard deviation normalization step incorporates illumination invariance, a step unaccounted for in Li et al. [3]. Thus two patches which differ only in absolute intensity values but are similar in the intensity variation pattern would be treated as similar by our models. The wider search space, compared to Ahmed et al. [2], gives invariance to large viewpoint variation. Further, performing inexact matching (correlation measure) over a wider search space gives us robustness to partial occlusions. Due to partial occlusions, a part(s) (P) of a person/object visible in one view may not be visible in others. Using a wider search space, our model looks for a part which is similar to the missing P within a wider neighborhood of P's original location. This is justified since typically adjacent regions of objects in an image have regularity/similarity in appearance. e.g. Bottom and upper half of torso. Now, since we are comparing two different parts due to the occlusion of P, we need to perform flexible matching. Thus, inexact matching is used.

**Cross Patch Feature Aggregation Layers**: The Normalized Correlation layer incorporates information from the local neighborhood of a pixel. We now seek to incorporate greater context information, to obtain a summarization effect. In order to do so, we perform 2 successive layers of convolution (with a ReLU filtered output) followed by pooling (by a factor of 2) of the output feature maps from the previous layer. We use $1 \times 1 \times 1500$ convolution kernels for the first layer and $3 \times 3 \times 25$ convolution kernels for the second convolution layer. Finally, we get 25 maps of dimension $5 \times 17$ each.

**Fully Connected Layers**: The fully connected layers collate information from pixels that are very far apart. The feature map outputs obtained from the previous layer are reshaped into one long 2125-dimensional vector. This vector is fed as input to a 500-node fully connected layer, which connects to another fully connected layer containing 2 softmax units. The first unit outputs the probability that the two images are same and the latter, the probability that the two are different.

One key advantage of the Normalized X-Corr model is that it has about half the number of parameters (about 1.121 million) as the model proposed by Ahmed et al. [2] (refer supplementary section for more details).

### 3.1.2 Fused Model

While the Normalized X-Corr model incorporates inexact matching over a wider search space to handle important challenges such as illumination variations, partial occlusions, or wide viewpoint changes, however it also suffers from occasional false matches. Upon investigation, we found that these false matches tended to recur, especially when the background of the false matches had a similar appearance to the person being matched (see supplementary). For such cases, an exact matching, such as taking a difference and constraining the search window might be beneficial. We therefore fuse the model proposed by Ahmed et al. [2] with Normalized X-Corr to obtain a Fused model, in anticipation that it incorporates the benefits of both models. Figure 3 shows a representative diagram.

We keep the tied convolution layers unchanged like before, then we fork off two separate pipelines: one for Normalized X-Corr and the other for Ahmed et. al.'s model [2]. The two separate pipelines output a 2125-dimensional vector each and then they are fused in a 1000-node fully connected layer. The outputs from the fully connected layer are then fed into a 2 unit softmax layer as before.

### 3.2 Training Algorithm

All the proposed architectures are trained using the Stochastic Gradient Descent (SGD) algorithm, as in Ahmed et al. [2]. The gradient computation is fairly simple except for the Normalized Correlation layer. Given two matrices, $E$ (from the first branch of the Siamese network) and $F$ (from the second branch of the Siamese network), represented by a N-dimensional vector each, the gradient pushed from the Normalized Correlation layer back to the convolution layers on the top branch is given by:

$$\frac{\partial normxcorr(E, F)}{\partial E_i} = \frac{1}{(N-1)\sigma_E} \left( \frac{F_i - \mu_F}{\sigma_F} - \frac{normxcorr(E, F)(E_i - \mu_E)}{\sigma_E} \right),$$

where $E_i$ is the $i^{th}$ element of the vector representing $E$ and other symbols have their usual meaning. Similar notation is used for the subnetwork at the bottom. The full derivation is available in the supplementary section.

## 4 Experiments

Table 1: Performance of different algorithms at ranks 1, 10, and 20 on CUHK03 Labeled (left) and CUHK03 Detected (right) Datasets.

| Method | r = 1 | r = 10 | r = 20 |
|---|---|---|---|
| **Fused Model (ours)** | **72.43** | **95.51** | **98.40** |
| **Norm X-Corr (ours)** | 64.73 | 92.77 | 96.78 |
| Ensembles [21] | 62.1 | 92.30 | 97.20 |
| LOMO+MLAPG [5] | 57.96 | 94.74 | 98.00 |
| Ahmed et al. [2] | 54.74 | 93.88 | 98.10 |
| LOMO+XQDA [22] | 52.20 | 92.14 | 96.25 |
| Li et al. [3] | 20.65 | 68.74 | 83.06 |
| KISSME [18] | 14.17 | 52.57 | 70.03 |
| LDML [14] | 13.51 | 52.13 | 70.81 |
| eSDC [25] | 8.76 | 38.28 | 53.44 |

| Method | r = 1 | r = 10 | r = 20 |
|---|---|---|---|
| **Fused Model (ours)** | **72.04** | **96.00** | **98.26** |
| **Norm X-Corr (ours)** | 67.13 | 94.49 | 97.66 |
| LOMO+MLAPG [5] | 51.15 | 92.05 | 96.90 |
| Ahmed et al. [2] | 44.96 | 83.47 | 93.15 |
| LOMO+XQDA [22] | 46.25 | 88.55 | 94.25 |
| Li et al. [3] | 19.89 | 64.79 | 81.14 |
| KISSME [18] | 11.70 | 48.08 | 64.86 |
| LDML [14] | 10.92 | 47.01 | 65.00 |
| eSDC [25] | 7.68 | 33.38 | 50.58 |

### 4.1 Datasets, Evaluation Protocol, Baseline Methods

We conducted experiments on the large CUHK03 dataset [3], the mid-sized CUHK01 Dataset [23], and the small QMUL GRID dataset [27]. The datasets are divided into training and test sets for our experiments. The goal of every algorithm is to rank images in the gallery image bank of the test set by their similarity to a probe image (which is also from the test set). To do so, they can exploit the training set, consisting of matched and unmatched image pairs. An oracle would always rank the ground truth match (from the gallery) in the first position. All our experiments are conducted in the single shot setting, i.e. there is exactly one image of every person in the gallery image bank and the results averaged over 10 test trials are reported using tables and Cumulative Matching Characteristics (CMC) Curves (see supplementary). For all our experiments, we use a momentum of 0.9, starting learning rate of 0.05, learning rate decay of $1 \times 10^{-4}$, weight decay of $5 \times 10^{-4}$. The implementation was done in a machine with NVIDIA Titan GPUs and the code was implemented using Torch and is available online [1]. We also conducted an ablation study, to further analyze the contribution of the individual components of our model.

**CUHK03 Dataset**: The CUHK03 dataset is a large collection of 13,164 images of 1360 people captured from 6 different surveillance cameras, with each person observed by 2 cameras with disjoint views [3]. The dataset comes with manual and algorithmically labeled pedestrian bounding boxes. In this work, we conduct experiments on both these sets. For our experiments, we follow the protocol used by Ahmed et al. [2] and randomly pick a set of 1260 identities for training and 100 for testing. We use 100 identities from the training set for validation. We compare the performance of both Normalized X-Corr and the Fused model with several baselines for both labeled [2, 3, 5, 14, 18, 21, 22, 25] and detected [2, 3, 5, 14, 18, 22, 25] sets. Of these, the comparison with Ahmed et al. [2] and with Li et al. [3] is of special interest to us since these are deep learning approaches as well. For our models, we use mini-batch sizes of 128 and train our models for about 200,000 iterations.

**CUHK01 Dataset**: The CUHK01 dataset is a mid-sized collection of 3,884 images of 971 people, with each person observed by 2 cameras with disjoint views [23]. There are 4 images of every identity. For our experiments, we follow the protocol used by Ahmed et al. [2] and conduct 2 sets of experiments with varying training set sizes. In the first, we randomly pick a set of 871 identities for training and 100 for testing, while in the second, 486 identities are used for testing and the rest for training. We compare the performance of both of our models with several baselines for both 100 test identities [2, 3, 14, 18, 25] and 486 test identities [2, 8, 9, 20, 21]. For our models, we use mini-batch sizes of 128 and train our models for about 50,000 iterations.

**QMUL GRID Dataset**: The QMUL underGround Re-Identification (GRID) dataset is a small and a very challenging dataset [27]. It is a collection of only 250 people captured from 2 views. Besides, the 2 images of every identity, there are 775 unmatched images, i.e. for these identities only 1 view is available. For our experiments, we follow the protocol used by Liao and Li [5]. We randomly pick a set of 125 identities (who have 2 views each) for training and leave the remaining 125 for testing. Additionally, the gallery image bank of the test is enriched with the 775 unmatched images. This makes the ranking task even more challenging. We compare the performance of both of our models with several baselines [11, 12, 15, 19, 22, 24]. For our models, we use mini-batch sizes of 128 and train our models for about 20,000 iterations.

Table 2: Performance of different algorithms at ranks 1, 10, and 20 on CUHK01 100 Test Ids (left) and 486 Test Ids (right) Datasets

| Method | r = 1 | r = 10 | r = 20 | Method | r = 1 | r = 10 | r = 20 |
|---|---|---|---|---|---|---|---|
| **Fused Model (ours)** | **81.23** | **97.39** | **98.60** | **Fused Model (ours)** | **65.04** | **89.76** | **94.49** |
| **Norm X-Corr (ours)** | 77.43 | 96.67 | 98.40 | **Norm X-Corr (ours)** | 60.17 | 86.26 | 91.47 |
| Ahmed et al. [2] | 65.00 | 93.12 | 97.20 | CPDL [8] | 59.5 | 89.70 | 93.10 |
| Li et al. [3] | 27.87 | 73.46 | 86.31 | Ensembles [21] | 51.9 | 83.00 | 89.40 |
| KISSME [18] | 29.40 | 72.43 | 86.07 | Ahmed et al. [2] | 47.50 | 80.00 | 87.44 |
| LDML [14] | 26.45 | 72.04 | 84.69 | Mirror-KFMA [20] | 40.40 | 75.3 | 84.10 |
| eSDC [25] | 22.84 | 57.67 | 69.84 | Mid-Level Filters [9] | 34.30 | 65.00 | 74.90 |

## 4.2 Training Strategies for the Neural Network

The large number of parameters of a deep neural network necessitate special training strategies [2]. In this work, we adopt 3 main strategies to train our model.

**Data Augmentation**: For almost all the datasets, the number of negative pairs far outnumbers the number of positive pairs in the training set. This poses a serious challenge to deep neural nets, which can overfit and get biased in the process. Further, the positive samples may not have all the variations likely to be encountered in a real scenario. We therefore, hallucinate positive pairs and enrich the training corpus, along the lines of Ahmed et al. [2]. For every image in the training set of size W×H, we sample 2 images for CUHK03 (5 images for CUHK01 & QMUL) around the original image center and apply 2D translations chosen from a uniform random distribution in the range of $[-0.05W, 0.05W] \times [-0.05H, 0.05H]$. We also augment the data with images reflected on a vertical mirror.

**Fine-Tuning**: For small datasets such as QMUL, training parameter-intensive models such as deep neural networks can be a significant challenge. One way to mitigate this issue is to fine-tune the model while training. We start with a model pre-trained on a large dataset such as CUHK01 with 871 training identities rather than an untrained model and then refine this pre-trained model by training on the small dataset, QMUL in our case. During fine-tuning, we use a learning rate of 0.001.

**Others**: Training deep neural networks is time taking. Therefore, to speed up the training, we implemented our code such that it spawns threads across multiple GPUs.

## 5 Results and Discussion

**CUHK03 Dataset**: Table 1 summarizes the results of the experiments on the CUHK03 Labeled dataset. Our Fused model outperforms the existing state-of-the-art models by a wide margin, of about 10% (about 72% vs. 62%) on rank-1 accuracy, while the Normalized X-Corr model gives a 3% gain. This serves as a promising response to our key research endeavor for an effective deep learning model for person re-identification. Further, both models are significantly better than Ahmed et al.'s model [2]. We surmise that this is because our models are more adept at handling variations in illumination, partial occlusion and viewpoint change.

Interestingly, we also note that the existing best performing system is a non-deep approach. This shows that designing an effective deep learning architecture is a fairly non-trivial task. Our models' performance, viz.-a-viz. non-deep methods, once again underscores the benefits of learning representations from the data rather than using hand-engineered ones. A visualization of some filter responses of our model, some of the result plots, and some ranked matching results may be found in the supplementary material.

Table 1 also presents the results on the CUHK03 Detected dataset. Here too, we see the superior performance of our models over the existing state-of-the-art baselines. Interestingly, here our models take a wider lead over the existing baselines (about 21%) and our models' performance rivals its own performance on the Labeled dataset. We hypothesize that incorporating a wider search space makes our models more robust to the challenges posed by images in which the person is not centered, such as the CUHK03 Detected dataset.

**CUHK01 Dataset**: Table 2 summarizes the results of the experiments on the CUHK01 dataset with 100 and 486 test identities. For the 486 test identity setting, our models were pre-trained on the training set of the larger CUHK03 Labeled dataset and then fine-tuned on the CUHK01-486 training set, owing to the paucity of training data. As the tables show, our models give us a gain of upto 16% over the existing state-of-the-art on the rank-1 accuracies.

**QMUL GRID Dataset**: QMUL GRID is a challenging dataset for person re-identification due to its small size and the additional 775 unmatched gallery images in the test set. This is evident from the low performances of

Table 3: Performance of different algorithms at ranks 1, 5, 10, and 20 on the QMUL GRID Dataset

| Method | r = 1 | r = 5 | r = 10 | r = 20 | Deep Learning Model |
|---|---|---|---|---|---|
| **Fused Model (ours)** | **19.20** | 38.40 | **53.6** | **66.4** | Yes |
| **Norm X-Corr (ours)** | 16.00 | 32.00 | 40.00 | 55.2 | Yes |
| KEPLER [24] | 18.40 | **39.12** | 50.24 | 61.44 | No |
| LOMO+XQDA [22] | 16.56 | 33.84 | 41.84 | 52.40 | No |
| PolyMap [11] | 16.30 | 35.80 | 46.00 | 57.60 | No |
| MtMCML [19] | 14.08 | 34.64 | 45.84 | 59.84 | No |
| MRank-RankSVM [15] | 12.24 | 27.84 | 36.32 | 46.56 | No |
| MRank-PRDC [15] | 11.12 | 26.08 | 35.76 | 46.56 | No |
| LCRML [12] | 10.68 | 25.76 | 35.04 | 46.48 | No |
| XQDA [22] | 10.48 | 28.08 | 38.64 | 52.56 | No |

existing state-of-the-art algorithms. In order to train our models on this small dataset, we start with a model trained on CUHK01 dataset with 100 test identities, then we fine-tune the models on the QMUL GRID training set. Table 3 summarizes the results of the experiments on the QMUL GRID dataset. Here too, our Fused model performs the best. Even though, our gain in rank-1 accuracy is a modest 1% but we believe this is significant for a challenging dataset like QMUL.

The ablation study across multiple datasets reveals that a wider search and inexact match each buy us at least 6% individually, in terms of performance. The supplementary presents these results in more detail and also compares the number of parameters across different models. Multi-GPU training, on the other hand gives us a 3x boost to training speed.

## 6 Conclusions and Future Work

In this work, we address the central research question of proposing simple yet effective deep-learning models for Person Re-Identification by proposing two new models. Our models are capable of handling the key challenges of illumination variations, partial occlusions and viewpoint invariance by incorporating inexact matching over a wider search space. Additionally, the proposed Normalized X-Corr model benefits from having fewer parameters than the state-of-the-art deep learning model. The Fused model, on the other hand, allows us to cut down on false matches resulting from a wide matching search space, yielding superior performance.

For future work, we intend to use the proposed Siamese architectures for other matching tasks in Vision such as content-based image retrieval. We also intend to explore the effect of incorporating more feature maps on performance.

## Footnotes

[1]https://github.com/InnovArul/personreid_normxcorr

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
