[Supplementary Material · Paper1373_supplementary-material-paper.pdf]

# Supplementary Material for the Paper: Deep Neural Networks with Inexact Matching for Person Re-Identification

**Arulkumar Subramaniam**
Indian Institute of Technology Madras
Chennai, India 600036
`aruls@cse.iitm.ac.in`

**Moitreya Chatterjee**
Indian Institute of Technology Madras
Chennai, India 600036
`metro.smiles@gmail.com`

**Anurag Mittal**
Indian Institute of Technology Madras
Chennai, India 600036
`amittal@cse.iitm.ac.in`

## 1 Overview

In this document, we present a derivation of the gradient, computed at the Normalized Correlation Layer of our models. We then present graphs that showcase the performance of our models in comparsion with other competing baselines on different datasets. Next, we present some examples to show how deftly our models handle challenges posed by partial occlusion, illumination variation, and viewpoint change. We also present prototypical examples of cases where our first model, Normalized X-Corr, obtains a false match, followed by examples of probe images where both of our proposed models fail. Finally, we present some convolution filter responses for instances where our model gets the right match at rank-1, followed by one where it fails to do so.

## 2 Gradient Computation at the Normalized Correlation Layer

Given two matrices, $X$ and $Y$, whose elements are arranged as two N-dimensional vectors, Normalized Correlation is given by:

$$normxcorr(X, Y) = \frac{\sum_{i=1}^{N}(X_i - \mu_X)(Y_i - \mu_Y)}{(N-1)\sigma_X \sigma_Y},$$

where $\mu_X, \mu_Y$ denotes the means of the elements of the 2 matrices X and Y respectively while $\sigma_X, \sigma_Y$ denotes their respective unbiased standard deviation (a small $\epsilon = 0.01$ is added to the unbiased standard deviation to avoid division by 0).

- The mean of a N-dimentional vector X, $\mu_X = \frac{\sum_{i=1}^{N} X_i}{N}$

- The unbiased standard deviation of a N-dimentional vector X, $\sigma_X = \sqrt{\frac{\sum_{i=1}^{N}(X_i - \mu_X)^2}{N-1}}$

Assume patch matrix $X$ comes from a response map of the top branch of the siamese network and $Y$ from a map of the bottom. We now compute the gradient with respect to the $i^{th}$ element of the $X$ as follows:

$$\frac{\partial normxcorr(X,Y)}{\partial X_i}$$

$$= \frac{1}{N-1} \left( \frac{(\sigma_X \sigma_Y) \left( \sum_{j=1,j\neq i}^{N} (\frac{-1}{N})(Y_j - \mu_Y) + (1 - \frac{1}{N})(Y_i - \mu_Y) \right) - \left( \sum_{i=1}^{N}(X_i - \mu_X)(Y_i - \mu_Y) \right) \left( \sigma_Y \frac{1}{2\sigma_X} \frac{1}{N-1} \left( \sum_{j=1,j\neq i}^{N} 2(X_j - \mu_X)\frac{-1}{N} + 2(X_i - \mu_X)(1 - \frac{1}{N}) \right) \right)}{(\sigma_X \sigma_Y)^2} \right)$$

(1)

$$\frac{\partial normxcorr(X,Y)}{\partial X_i}$$

$$= \frac{1}{N-1} \left( \frac{(\sigma_X . \sigma_Y) \left( (Y_i - \mu_Y) - \sum_{j=1,j\neq i}^{N} (\frac{1}{N})(Y_j - \mu_Y) - \frac{1}{N}(Y_i - \mu_Y) \right) - \left( \frac{\sum_{i=1}^{N}(X_i - \mu_X)(Y_i - \mu_Y)}{(N-1)\sigma_X} \right) \left( \sigma_Y \left( (X_i - \mu_X) - \sum_{j=1,j\neq i}^{N} \frac{1}{N}(X_j - \mu_X) - \frac{1}{N}(X_i - \mu_X) \right) \right)}{(\sigma_X \sigma_Y)^2} \right)$$

(2)

By substituting,

$$\frac{\sum_{i=1}^{N}(X_i - \mu_X)(Y_i - \mu_Y)}{(N-1)\sigma_X} = \sigma_Y normxcorr(X,Y)$$

We get,

$$\frac{\partial normxcorr(X,Y)}{\partial X_i}$$

$$= \frac{1}{N-1} \left( \frac{(\sigma_X . \sigma_Y) \left( (Y_i - \mu_Y) - \frac{1}{N}\sum_{i=1}^{N}(Y_i - \mu_Y) \right) - \left( \sigma_Y^2 normxcorr(X,Y) \left( (X_i - \mu_X) - \frac{1}{N}\sum_{i=1}^{N}(X_i - \mu_X) \right) \right)}{(\sigma_X \sigma_Y)^2} \right)$$

(3)

Now since,

$$\frac{1}{N}\sum_{i=1}^{N}(Y_i - \mu_Y) = \mu_Y - \mu_Y = 0$$

We have,

$$\frac{\partial normxcorr(X,Y)}{\partial X_i}$$

$$= \frac{1}{N-1}\left(\frac{(\sigma_X.\sigma_Y)(Y_i-\mu_Y) - \left(\sigma_Y^2 normxcorr(X,Y)(X_i-\mu_X)\right)}{(\sigma_X.\sigma_Y)^2}\right) \quad (4)$$

$$= \frac{1}{(N-1)}\left(\frac{Y_i-\mu_Y}{\sigma_X\sigma_Y} - \frac{normxcorr(X,Y)(X_i-\mu_X)}{\sigma_X^2}\right) \quad (5)$$

$$= \frac{1}{(N-1)\sigma_X}\left(\frac{Y_i-\mu_Y}{\sigma_Y} - \frac{normxcorr(X,Y)(X_i-\mu_X)}{\sigma_X}\right) \quad (6)$$

Equation 6 represents the gradients that are then pushed back to the top branch of the siamese network.

The gradient for the bottom branch, can similarly be computed with respect to the elements of $Y$ as follows:

$$\frac{\partial normxcorr(X,Y)}{\partial Y_i} = \frac{1}{(N-1)\sigma_Y}\left(\frac{X_i-\mu_X}{\sigma_X} - \frac{normxcorr(X,Y)(Y_i-\mu_Y)}{\sigma_Y}\right) \quad (7)$$

## 3 Models comparison

### 3.1 Ablation study

The ablation study conducted on different datasets of various sizes gives the following results:

| Title | NormXcorr, 5x5 search | | | NormXcorr, 5x12 search | | | [1], 5x5 search | | | [1], 5x12 search | | |
|---|---|---|---|---|---|---|---|---|---|---|---|---|
| | r = 1 | r = 10 | r = 50 | r = 1 | r = 10 | r = 50 | r = 1 | r = 10 | r = 50 | r = 1 | r = 10 | r = 50 |
| CUHK03 labeled | 62.43 | 92.22 | 99.6 | **64.73** | 92.77 | 99.6 | 54.74 | **93.3** | **99.7** | 57.60 | 90.63 | 99.17 |
| CUHK03 detected | 63.12 | 92.76 | 99.2 | **67.13** | **94.49** | **99.73** | 44.96 | 83.47 | 99.4 | 54.31 | 90.24 | 99.18 |
| CUHK01 100 tests | 72.3 | 95.8 | 99.6 | **77.43** | **96.67** | 99.29 | 65.0 | 94.0 | **99.9** | 69.7 | 95.03 | 99.13 |
| CUHK01 486 tests | 56.79 | 84.43 | 95.95 | **60.17** | **86.26** | **96.44** | 47.5 | 80.25 | 96.30 | 49.31 | 81.48 | 95.95 |

### 3.2 Parameter complexity

| | #parameters |
|---|---|
| [1]ś model | 2,308,147 |
| NormXcorr model(ours) | 1,121,222 |
| Fused model(ours) | 2,222,147 |

The proposed NormXcorr model has half the parameters as that of [1]. In [1], the majority of parameters ($\sim 97\%$) are due to large number of inputs to every node of the Fully Connected (FC) Layer ([2 branches x 25 maps x 18 height of a map x 5 width of a map x 500 FC nodes]). Since NormXcorr layer is symmetric, our model has only a single branch which reduces the parameters of Fully-connected layer by half ([25 maps x 17 height x 5 width x 500 FC nodes]). This also compensates for the minor increase in parameters due to wider search area over [1] ([1500 maps resulting from NormXcorr model – 1250 maps resulting from [1]] = 250 maps). In fused model, the fully connected layer nodes of the state-of-the-art model [1] are kept as 500 (instead of 1000 from [1]) to reduce the parameters needed in Fully-connected layers.

# 4 Performance Plots of Different Algorithms

In this section, we present a comparison of the performances of our models, viz.-a-viz. other competing baselines on different datasets. The performance is represented by Cumulative Matching Characteristic Curves (CMC Curve), where the y-axis represents what percentage of the probe images, in the test, that have been correctly matched to the gallery, for a particular number of retrievals, and the X-axis represents the ranks, i.e. the number of retrievals.

(a) CUHK03 labeled dataset

(b) CUHK03 detected dataset

(c) CUHK01 - 100 test identities

(d) CUHK01 - 486 test identities

(e) QMULGRID

Figure 1: CMC Curves (best viewed in color) for competing algorithms on different datasets.

# 5  Sample Results

In this section, we present a comparison of our models with that of Ahmed et al. [1] for person re-identification under various settings.

Figure 2 presents an instance where partial occlusion is taken care of by our models.

Figure 2: The 3 rows of images in the figure, correspond to retrieval results by Ahmed et al. [1], Normalized X-Corr, and the Fused Model respectively (first to third). The first column of the figure represents a probe image, while the next 20 columns represent the top-20 retrieved results for the 3 models. The ground truth match is highlighted with a green bounding box, all other gallery images have red text on them.

Our models also take care of illumination variation as is evident in Figure 3.

Figure 3: The 3 rows of images in the figure, correspond to retrieval results by Ahmed et al. [1], Normalized X-Corr, and the Fused Model respectively (first to third). The first column of the figure represents a probe image, while the next 20 columns represent the top-20 retrieved results for the 3 models. The ground truth match is highlighted with a green bounding box, all other gallery images have red text on them.

Large viewpoint variations are also handled by our models. Figure 4 presents an example.

Figure 4: The 3 rows of images in the figure, correspond to retrieval results by Ahmed et al. [1], Normalized X-Corr, and the Fused Model respectively (first to third). The first column of the figure represents a probe image, while the next 20 columns represent the top-20 retrieved results for the 3 models. The ground truth match is highlighted with a green bounding box, all other gallery images have red text on them.

As has been discussed in the paper, the Normalized X-Corr model does suffer from occasional false matches, for instance in case of the probe image in Figure 5.

Figure 5: The 3 rows of images in the figure, correspond to retrieval results by Ahmed et al. [1], Normalized X-Corr, and the Fused Model respectively (first to third). The first column of the figure represents a probe image, while the next 20 columns represent the top-20 retrieved results for the 3 models. The ground truth match is highlighted with a green bounding box, all other gallery images have red text on them.

Figure 6 presents a probe image from the test set of CUHK03 Labeled Dataset where all models (including the baseline approach) fails to get the right match in the top-20 retrieved results.

Figure 6: The 3 rows of images in the figure, correspond to retrieval results by Ahmed et al. [1], Normalized X-Corr, and the Fused Model respectively (first to third). The first column of the figure represents a probe image, while the next 20 columns represent the top-20 retrieved results for the 3 models. All false matches from the gallery, have red text on them.

Finally, Figure 7 presents the output feature maps after the second stage of convolution, for a probe image for which both of our models get the correct match at rank-1. Figure 8 presents the output feature maps after the second stage of convolution, for a probe, where Normalized X-Corr fails to get the correct match at rank-1 but the Fused model gets it right at the first rank.

Figure 7: All the images in the figure correspond to the output obtained after the second stage of convolution under different settings. Columns 2 through 26 represent the 25 output feature maps under different settings, while column 1 shows the original image. The first 3 rows represent the processed output of the probe image (first - Ahmed et al. [1], second - Normalized X-Corr, third - Fused Model), the next 3 rows represent the rank-1 image outputs for the different models (fourth - Ahmed et al. [1], fifth - Normalized X-Corr, sixth - Fused Model), the last 3 rows represent the ground truth match-image outputs for the different models (seventh - Ahmed et al. [1], eighth - Normalized X-Corr, ninth - Fused Model).

Figure 8: All the images in the figure correspond to the output obtained after the second stage of convolution under different settings. Columns 2 through 26 represent the 25 output feature maps under different settings, while column 1 shows the original image. The first 3 rows represent the processed output of the probe image (first - Ahmed et al. [1], second - Normalized X-Corr, third - Fused Model), the next 3 rows represent the rank-1 image outputs for the different models (fourth - Ahmed et al. [1], fifth - Normalized X-Corr, sixth - Fused Model), the last 3 rows represent the ground truth match-image outputs for the different models (seventh - Ahmed et al. [1], eighth - Normalized X-Corr, ninth - Fused Model).

# References

[1] Ejaz Ahmed, Michael Jones, and Tim K Marks. An improved deep learning architecture for person re-identification. In *Proceedings of the IEEE Conference on Computer Vision and Pattern Recognition*, pages 3908–3916, 2015.