[Reviews · NeurIPS 2016]

Reviewer 1

Summary

The paper addressed the re-identification problems, I.e. identifying two images of the same person as belonging to the same man/woman. A Siamese CNN architecture is suggested, with novel dedicated layer performing normalized correlation of each location with a large set of neighboring locations. The new layer is embedded in the middle of a standard CNN. The empirical results how a significant advantage of the new architecture over previous ones, especially when combined in a fusion with a more standard approach.

Qualitative Assessment

The pros and Cons of the paper are as follows: + the empirical results are strong, showing consistent advantage of the proposed method over previous art (as far as I can judge – I am not familiar with re-identification literature) + the new normalized correlation layer is a sensible architecture for the re-identification task. Each input map is compared alone using normalized correlation to a large array of might-ne-relevant positions, providing reach output (1500 output neurons per location – that’s a lot) for further processing. - The domain of re-identification is a relatively small one, of limited interest (hence significance of the paper is low, as it is relevant to a relatively small community) - clarity of presentation is low, with English levels which should be considerably improved. More detailed comments: Page 1: Lines 50-53: I did not understand the argument in these lines. It is stated that due to a large search area and inexact matching object parts in image 1 may be matched to background parts in image 2, and this is stated as a remedy to the partial occlusion problem. However, if this happens without significant penalty, irrelevant matches contribute to the score and may come to dominate it if there are many of them. - This argument seems to repeat in lines 155-158, and again I do not understand it Page 3: Line 77: “…learning the best representation” – this sentence is a too-bold claim. CNNs are not guaranteed, and are not, learning the best representation, and it is not even well defined what the ‘best representation’ is. Page 4: Line 134: - Why does ‘a’ run in 1..12 and not in 3…10? I mean, a represents the center of a 5*5 neighborhood and the map size is 12*37, so only y values of 3..10 correspond to patches inside the image. Is padding used? - Why is y treated different than x in this formalism, that is: y is searched exhaustively , while x is searched only locally. Is it because the typical long-but-narrow aspect ratio of standing humans? Page 5: Line 168: why does the output needs two units when the output is a single binary variable? A simple sigmoid can be used here instead of the exploss. Section 3.2: I checked the derivative formula given and found it correct. Page 6, table 1: it is not stated clearly enough what the measurement score is. I assume it is the percentage of correct retreivals up to rank r, right? This should eb explicitly explained. Page 7-8: the experimental results show a significant advantage of the suggested method. The English level is not high enough, where sometimes the words choice make reading harder than it should be. Some examples Abstract: I believe ‘upto’ is not a word Line 30: ‘entails’ is probably not the right word here. Maybe ‘is’ Line 35: “Have’ -> ‘contain’ Line 58” “show Normalized X-Corr to hold promise” -> “show Normalized X-Corr to fulfill the promise” Line 102-103: “Besides, Normalized X-Corr stands out by retaining the matching outcome, corresponding to every candidate in the search space of a pixel rather than choosing only the maximum match” – I did not understand this sentence.

Confidence in this Review

2-Confident (read it all; understood it all reasonably well)


Reviewer 2

Summary

This paper proposes two CNN models for person re-identification. The novel aspects of the first model are normalized cross correlation (Normalized X-Corr) layer and wider search area for patch matching. The second model is the combination of the Ahmed’s model [1] and the first model. The experimental results show the substantial performance gain over the state-of-the-art deep learning based methods on three benchmark datasets.

Qualitative Assessment

I think the proposed method seems to be somewhat novel and the performance is impressive. However, the evaluation and the writing are clearly rushed and incomplete. Detailed comments are bellows: Pros ) The performance of the proposed method is impressive. Even with only Normalized X-Corr model, the proposed method achieves substantially higher identification rates on CUHK01 and CUHK03 datasets than state-of-the arts. The Normalized X-Corr layer is interesting and seems to reasonable if the performance is really gained by this layer (see Cons. comment). This paper shows that combination of the different deep architectures improves the performance. This is simple idea, but as far as I know, this is not explored except a work [A]. I consider that the work [A] is published after the submission of this paper. [A] F.Wang, W.Zuo, L.Lin, D.Zhang, L.Zhang, Joint Learning of Single-image and Cross-image Representations for Person Re-identification, In IEEE Conference on Computer Vision and Pattern Recognition (CVPR), pp.1288-1296, 2016. Cons ) The main drawback of this paper is the incomplete baseline comparison. The proposed method has two differences from the Ahmed et al [1]. These are; (1) Normalized X-Corr layer and (2) wider area search. However, this paper does not report the baseline results except the reported results by Ahmed et al. [1]. Therefore, contributions of each part ((1) and (2)) are not understand. For example, the high performance of the proposed method might be comes from the wider area search and the Normalized X-Corr layer might be not contributed to the performance, and vice versa. The authors could compare the performance with the Cross Input Neighborhood layer of Ahmed’s model with wider search area, and Normalized X-Corr layer with smaller area search of Ahmed’s model, for example. One advantage claimed in this paper “the parameters of the Normalized X-Corr model is about half number compared to the Ahmed’s model (eg. line170-171)” is incorrect. In Fig.3, the size of Normalized X-Corr layer is 12 x 37 x 1500 while the Cross Input Neighborhood layers of Ahmed’s model is 12 x 37 x 1250. Clearly, the parameters of Ahmed’s model is smaller. This paper doesn’t explain how the number of parameters comes from and why it is smaller than that of Ahmed’s model. In my understanding, the depth of Cross Input Neighborhood is comes from 5x5 (search area) x 2 (asymmetric distance) x 25 (feature maps) = 1250. While the depth of Normalized X-Corr is comes from 12x5 (search area) x 1 (symmetric distance) x 25 (feature maps) = 1500. Due to the large search area, the parameters of the proposed model is larger than that of Ahmed’s model. The VIPeR dataset is the most commonly used dataset for person re-identification. However, the experimental results on this dataset is not reported. It would be better to report the results on VIPeR dataset. Writing quality of the paper is low: - Line 21. and 59: What is jumped and which datasets? - Line 29: “1.1 Problem Definition” can be removed. - Line 51, 158: What do you mean “images are regular” ? - Line 154: It would be better to explain how did Ahmed et al. set search space. - Line 183 “they are fused in a 1000 node fully connected layer”: Is it two separate 500 node fully connected layers (Fig. 3) ? - Line 250: How much did you get speed up by multi-GPU training? - Line 255, “Normalized X-Corr model give a 3% gain”: It should be “1.63% gain”. (64.73% vs 62.1%). - Line 275, “a gain of upto 16% over the existing state-of-the art”: Please specify which setting on the CUHK01. - Some capitalization should be collected or unified. eg., “Person Re-Identification” → “person re-identification” “Dataset” → “dataset” “Underground Re-identification (GRID)” →“underGround Re-IDenditifcaiton (GRID)” Supplementary material: - How did you obtain Eq.(1) ? More explanations are need. - Please use \eqnarray (if you use latex) for Eq.(1-2) and Eq.(5-7) . - Eq.(3) should be removed. - I do not understand your intention to show the Fig.7-8.

Confidence in this Review

3-Expert (read the paper in detail, know the area, quite certain of my opinion)


Reviewer 3

Summary

The author proposes a Normalized Correlation layer based on Ahmed's work to better deal with large viewpoint variation problem. Experiments show that the proposed method can achieve much better result when combined with the original Ahmed's network. This paper simply extends Ahmed's work and the idea is straightforward.

Qualitative Assessment

This paper focuses on person re-identification problem and extends a previous Ahmed's CVPR 2015 paper. The idea is rather straightforward. In Ahmed's work, the Cross-Input Neighborhood Differences layer can only capture relatively local feature correspondences. However, in real cases, large viewpoint variation problem could not be handled well by the Cross-Input Neighborhood Differences layer. So the author proposes a Normalized Correlation layer to better deal with large viewpoint variation problem as well as illumination and occlusion. The idea is rather simple and straightforward, i.e. search exhaustively along the x axis (width), and search in a small range [-2,2] along y axis (height). There are much to be done in the task of person re-identification. How to align different parts of two persons in different viewpoints. Will pose estimation help a lot? Personally, I feel that human make decisions by looking at different local parts such as face, gender, pattern on clothes, color of clothes, accessories and so on. If this is true, carefully design that involving detection, alignment, spatial transformer net and so on may further improve person re-identification result by a large margin. So my suggestion is that the authors should try to solve the problem to make it practical instead of doing some easy improvement. And as far as I know, one paper in CVPR 2016[1] and several unpublished papers[2] can achieve similar results compared to this paper. Overall, the paper implements a new layer, and improves person Re-ID result by collecting far-away feature correspondences in x-axis. But the contribution relatively limited. [1] Learning Deep Feature Representations With Domain Guided Dropout for Person Re-Identification, Tong Xiao, Hongsheng Li, Wanli Ouyang, Xiaogang Wang. CVPR 2016. [2] End-to-End Comparative Attention Networks for Person Re-identification, Hao Liu, Jiashi Feng, Meibin Qi, Jianguo Jiang and Shuicheng Yan. http://arxiv.org/pdf/1606.04404.pdf

Confidence in this Review

2-Confident (read it all; understood it all reasonably well)


Reviewer 4

Summary

The paper proposes two new deep networks for person re-identification. The first one has a layer of normalized correlation which captures the similarity of two input images. The second one is a variation of the first one. This network has one more branch containing a new layer of which incorporating information from the local neighborhood of a pixel.

Qualitative Assessment

Person re-identification is very useful in application. The proposed solution in this paper is reasonable. The two new additional layers are helpful. And the computation details of the normalized correlation are given in the supplementary. From the experiment results, we can see that the Nom X-corr model is worse than previous methods on the dataset of QMUL GRID. By contrast, the fuse model gains the best performance. However, in the section of training algorithm, only the gradient is presented. The fused model has a new branch which is different from [1]. Therefore, I don’t think the training method of [1] can be directly used. How to jointly training the two branches? The paper should give proper description. This is critical point for a mixed network. Moreover, a discussion on complexity of the method should be included in the experiment.

Confidence in this Review

2-Confident (read it all; understood it all reasonably well)


Reviewer 5

Summary

The authors propose two DCNN architectures for person re-identification that allow inexact matching to handle variations in data. Experiments on multiple benchmark datasets are done to evaluate the proposed algorithms and compare them to published appraoches.

Qualitative Assessment

This paper presents a CNN architecture for person re-identification with improvements of marginal novelty on [1] by 1) using a symmetric normalized cross correlation (versus differences) to match intermediate representations of two input images, 2) searching a larger area and feeding all matching scores to next level CNN processing (versus keeping only the best scores), and 3) using more convolution layers. In a second architecture the proposed CNN is fused with the architecture in [1] to further improve the performance. What is the most impressive is that there is significant performance gain using the fused architecture on three benchmark datasets, especially for the CUHK03 dataset. As the paper is inspired by [1], I suggest to give a more detailed description of [1] is Sec. 2.2 and highlight what is in common and what is different with [1] in Sec. 3. Other than this the paper is well organized with sufficient references and reads well.

Confidence in this Review

2-Confident (read it all; understood it all reasonably well)


Reviewer 6

Summary

This paper proposes a deep end-to-end network using a “siamese” type architecture by extending the work by Ahmed et. Al. [1] to address multi-camera person re-identification. Specifically, it replaces the cross input neighborhood difference layer in [1] with a Normalized Correlation layer to exploit the strength of correlation analysis which is known to handle illumination variation well in other areas of computer vision including person re-identification. A clever use of wider search space (compared to [1]) to find neighborhood patches from other view enabled the method to address large viewpoint variations which in turn made the proposed approach more resistant to occlusion. In addition, the two stream network which is a result of fusing [1] with the proposed method shows remarkable boost in benchmark datasets. The proposed method is evaluated on 3 benchmark datasets.

Qualitative Assessment

Strong points: 1. The use of normalized correlation in place of cross input neighborhood difference layer as in [1] facilitates illumination invariance 2. The use of wider search space for neighborhood patches makes the model fit for occlusion robustness 3. Experimental results on 3 benchmark datasets show impressive improvements over state-of-the-art. Confusing points: 1. I’m not sure if the “fused” network can be claimed as a separate CNN-based architecture (as said in line 9 ‘two’ CNN based architecture). It merely uses the same architecture as [1] in one stream throughout and fuses the proposed architecture at the end. 2. The paper says about using around half the parameters compared to [1]. An explanation about this difference would be helpful. On an intuitive level, is it because the number of feature maps/channels in the last few layers (between correlation layer and the final binary output layer) being half (25) compared to [1] (50)? A related question may be – how does the use of wider search space and hence computing correlation with a lot more number of neighborhood patches scale the computational load (say with [1])? 3. The experimental results are quite impressive. However, it is not clear whether the use of wider search space or the use of normalized cross correlation features have boosted the performance so much. A comparison with an extension of [1] with similar wider search space would have given an insight towards this. However, the lack of this experiment does not affect my view towards this paper as this addition can be kept as a future work. Finally, a suggestion to include some qualitative results in the main paper. Many a time, the re-identification papers don’t provide qualitative results and this does not help in supporting some broad claims (like occlusion robustness, illumination invariance etc.). However, the supplementary material has some impressive results on showing occlusion robustness or viewpoint variation. Inclusion of some of these (if space permits) would greatly benefit the paper and may encourage the community to include them subsequently. So, even though the paper’s count towards ‘novelty’ poses some question in my mind and the design of the new layer seems a little ad-hoc, I’d recommend it as a poster (at this moment). The experimental results seem to be a big plus point for this paper and this would encourage the community to look further into its strength and weaknesses and try to use it in other areas of computer vision and machine learning as well.

Confidence in this Review

3-Expert (read the paper in detail, know the area, quite certain of my opinion)